# Reversing Years for Global Food Security: A Review of the Food Security Situation in Sub-Saharan Africa (SSA)

**DOI:** 10.3390/ijerph192214836

**Published:** 2022-11-11

**Authors:** Abdulazeez Hudu Wudil, Muhammad Usman, Joanna Rosak-Szyrocka, Ladislav Pilař, Mortala Boye

**Affiliations:** 1Department of Agricultural Economics and Extension, Federal University Dutse, Dutse 720101, Nigeria; 2Institute of Agricultural and Resource Economics, University of Agriculture Faisalabad, Faisalabad 38000, Pakistan; 3Department of Production Engineering and Safety, Faculty of Management, Czestochowa University of Technology, 42-201 Częstochowa, Poland; 4Department of Management, Czech University of Life Sciences Prague, 16500 Prague, Czech Republic; 5School of Agriculture and Environment Sciences, University of the Gambia, Kanifing P.O. Box 3530, The Gambia

**Keywords:** food security, SDG2, inequality, policy, Sub-Saharan Africa

## Abstract

All around the world, inequalities persist in the complex web of social, economic, and ecological factors that mediate food security outcomes at different human and institutional scales. There have been rapid and continuous improvements in agricultural productivity and better food security in many regions of the world during the past 50 years due to an expansion in crop area, irrigation, and supportive policy and institutional initiatives. However, in Sub-Saharan Africa, the situation is inverted. Statistics show that food insecurity has risen since 2015 in Sub-Saharan African countries, and the situation has worsened owing to the Ukraine conflict and the ongoing implications of the COVID-19 threat. This review looks into multidimensional challenges to achieving the SDG2 goal of “End hunger, achieve food security and improved nutrition, and promote sustainable agriculture” in Sub-Saharan Africa and the prosper policy recommendations for action. Findings indicate that weak economic growth, gender inequality, high inflation, low crop productivity, low investment in irrigated agriculture and research, climate change, high population growth, poor policy frameworks, weak infrastructural development, and corruption are the major hurdles in the sustaining food security in Sub-Saharan Africa. Promoting investments in agricultural infrastructure and extension services together with implementing policies targeted at enhancing the households’ purchasing power, especially those in rural regions, appear to be essential drivers for improving both food availability and food access.

## 1. Introduction

The effort to eradicate hunger, food insecurity, and other forms of malnutrition has been the topic of many studies [1]. Nevertheless, despite recent advances in these areas, many nations, particularly in developing countries like Sub-Saharan Africa, still face significant food insecurity challenges [1,2]. In the mid-1970s, food insecurity prompted world leaders to acknowledge for the first time their shared duty to end hunger and malnutrition [3]. Nonetheless, in the 48 least developed countries, per capita food consumption fell between 1980 and 1998, while it rose in most of the developing world [4]. According to the 1996 World Food Summit (WFS), the number of hungry people worldwide was predicted to fall by at least 20 million every year between 2000 and 2015. Even though some areas made great strides in the 20 years before the year 2000, the latest data on the number of undernourished people around the world shows that the average annual decrease since the 1996 WFS has been only 2.5 million, which is far below the level needed to reach the WFS goal of significantly reducing the level of undernourishment by 2015 [5].

During the past three decades, global food production has increased faster than the population growth rate. However, in 2018–2019, more than 830 million people worldwide, mainly in South Asia and Sub-Saharan Africa, experienced physical food insecurity; this figure does not account for obesity or micronutrient deficiency, which affect more than a billion people.

Stable distribution networks are critical to food security. Since the beginning of 2020, the COVID-19 epidemic has been putting this stability through one of the most severe tests it has ever experienced. As evidenced by lockdowns, economic downturns, trade restrictions, and soaring food prices, the pandemic has become more than just a health issue. It is becoming a significant economic threat to world food security [6]. Food costs have hit all-time highs due to the war in the Ukraine, food supply issues, and the ongoing economic impacts of the COVID-19 epidemic [7]. People in poor and middle-income countries spend a large portion of their income on food, leaving them particularly vulnerable to price increases.

Despite years of persistent efforts to increase the production and quality of the world’s food resources, food insecurity still exists, particularly in African countries. The information on Sub-Saharan African Food Security (SSAFB) estimates that one in three persons in the region is malnourished [8]. It will be difficult for Africa to have sustainable economic development if its people are malnourished and unwell. Many African nations continue to struggle with undernourishment despite receiving food aid from international organizations. Due to changes in consumer habits, population growth, and global economic upheavals, the food security situation in Sub-Saharan Africa (SSA) is deteriorating [9]. Sub-Saharan Africa (SSA) has seen a large number of economic changes. No matter how hard people work, the economy is still shrinking and endangering everyone’s way of life, especially those in rural areas. Since 2015, there has been an increase in food insecurity in the SSA. The SDG2 of the 2030 Agenda for Sustainable Development is to end world hunger and malnutrition. Africa, however, is not on course to achieve this objective.

A plethora of studies has pointed to several factors that contribute significantly to food insecurity in Africa. For instance, low economic growth [10], gender inequality [11], food price inflation, low agricultural productivity, drought [12], inadequate investment in irrigation [9], climate change [13,14], and rapid population growth [15,16]. Floods, severe rainfall, drought, cyclones, and storms are all climate-induced phenomena that negatively influence rural income and food security in low-income developing countries [16,17,18]. 

Similarly, ref. [19] outlined the impact of poor social inclusion, ref. [20] emphasized the improper use of natural resources, and [21] explored the impact of politics and political instability The present study, however, views food insecurity in Africa from a multidimensional perspective; not only focusing on the determinants of food insecurity in the SSA, but also exploring remedies for SSA’s ongoing food security dilemma. The following questions are therefore addressed in this review:Why do SSA countries respond slowly to the SDG2 goal of “End hunger, achieve food security and improved nutrition, and promote sustainable agriculture”?What strategy should Sub-Saharan African countries adopt to improve food security sustainably?

While addressing these questions, this study makes significant contributions. First, the study contributes to the food security literature by assessing the food security situation in Sub-Saharan Africa. Secondly, the study contributes to the sustainable development goal by identifying and analyzing the challenges for attaining the SDG2 of ending hunger, food insecurity, and sustainable agriculture in Africa. Third, the paper serves as a valuable resource for academics interested in food and nutrition security, illuminating the most recent findings and encouraging them to build upon previous investigations and anticipate future developments.

The rest of the article proceeds as follows. The following section focuses on an overview of the food security concept, followed by the global food insecurity trend. Subsequently, the article presents an overview of the global hunger index in Sub-Saharan Africa. The fourth section delves into the drivers of food insecurity in Sub-Saharan Africa. Finally, the conclusions and recommendations are presented.

### 1.1. An Overview of the Concept of Food Security

Since the 1974 World Food Conference, at which concerns of hunger, famine, and the food crisis were explored at length, the complex idea of food security has garnered continuous attention and economic relevance [22,23,24]. Although it has evolved, the concept of food security remains the same: “a situation that exists when all people, at all times, have physical, social, and economic access to sufficient, safe, and nutritious food that meets their dietary needs and food preferences for an active, healthy life.” This multifaceted definition is based on four pillars: Availability (or adequate food supply), affordability (or low prices), stability (no food shortages or seasonal swings), and utilization [25,26]. 

Webb and Sheeran [27] claimed that individuals go hungry not because food is not readily available in the market, but because they cannot afford to buy it. Therefore, it is essential to differentiate between food supply and food demand. Availability is a term commonly used to describe the state of food supplies, and thanks to developments in agricultural output, availability has increased over the previous few decades. The unequal distribution of food occurs not only within households, but the term “access” is also usually employed to highlight the issue of demand [28]. The global food price issue could reverse the decades of progress in alleviating hunger and malnutrition and putting an end to extreme poverty.

#### 1.1.1. The Trend of the Global Food Insecurity Situation since 2000

In 2017, about 1.9 billion individuals lacked consistent access to appropriate nutrition [29]. While most of those affected by food insecurity reside in the low-income countries in Sub-Saharan Africa and South Asia, this issue impacts individuals worldwide [30]. All nations must be able to acquire and maintain nutrient-dense diets that are diverse. Achieving food security and eliminating hunger are two 2030 Sustainable Development Goals of the United Nations. Despite the world’s nations concentrated efforts to alleviate global hunger, the number of undernourished people grew in 2015, following a decline from the 2003 to 2014 (Figure 1). According to the Food and Agriculture Organization (FAO), up to the year 2020, over 825 million people will have gone hungry, and roughly 2 billion will have been moderately or severely food insecure [31]. Due to the additional 60 million victims of hunger since 2014, the number of undernourished people is projected to reach 840 million by 2030 [30], which is a significant rise over the present estimated figure of 380 million [30] (Figure 1). Every day, a colossal 830 million people worldwide suffer from hunger (Figure 1). As a direct result of COVID-19 and the ongoing conflict in the Ukraine, the number of people experiencing acute food insecurity rose dramatically from 135 million in 2019 to 345 million in 2022 [32]. Even with the recovery of the global economy, [33] anticipated that eight percent of the world’s population, or 670 million people, would still be hungry in 2030. This is comparable to 2015 when the 2030 Agenda for Sustainable Development established a goal date for this decade to abolish world hunger, food insecurity, and malnutrition. According to the World Food Program, 50 million people in 45 countries are at risk of extreme hunger [32]. Historically, droughts and other natural disasters have been the leading causes of food scarcity [34].

#### 1.1.2. Sub-Saharan Africa and the Global Hunger Index 

The International Food Policy Research Institute (IFPRI) developed a scale to assess food insecurity called the Global Hunger Index (GHI), which takes into account three indicators: stunting, malnutrition, and infant mortality [29]. On a scale from 0 (no hunger) to 100 (the worst case), this is measured with the following criteria. Child wasting: the number of children under five who are too thin for their height; this is a sign of severe malnutrition. Child stunting: the number of children under five who are stunted, which means they are short for their age because they do not get enough food. Anemia: the percentage of the population with anemia, which shows how many people are not getting enough calories (partially reflecting the serious interaction of inadequate nutrition and unhealthy environments). 

South Asia and Africa, south of the Sahara, have the highest levels of hunger in the world according to the GHI values, 27.4 and 27.1, respectively, in the year 2022 (Figure 2), and both regions are experiencing a stagnation in the fight against hunger. In South Asia and SSA, where the scores were 28.0 and 28.1 in 2014, progress in eliminating hunger has virtually stagnated in comparison to the progress recorded from the year 2000 to 2014 (Figure 2). In terms of the development required to meet the second Sustainable Development Goal of “Zero Hunger” by 2030, Africa, south of the Sahara, is gravely behind the agenda [35].

## 2. The Sub-Saharan African Food Security Situation

Africa is home to at least one-third of the world’s undernourished population and is the only continent where agricultural productivity per capita has been falling for the past 30 years [36]. Africa’s population is expected to expand from its estimated 1.2 billion in 2020 to over 1.8 billion by 2050; more than half of the continent’s inhabitants are under 20 [37]. The agricultural sector employs around 60% of the labor force, produces approximately 22% of the GDP, and accounts for more than 10% of export revenues. The SSA countries coverage in the international food security assessment in this study is presented in Figure 3.

According to research conducted by Harvard University Professor Calestous Juma, Africa could achieve food self-sufficiency if it stopped depending on food imports [9]. Farsund, Daugbjerg, and Langhelle [38] concluded that Africa’s land resource could quickly produce an additional 100 million tons of grain-equivalent a year if it were intensively farmed. Africa has significant untapped potential, as seen by the region’s relatively low yields compared to other regions with similar agro–ecological zones like South Asia. Utuk and Daniel [39] reported that about 800 million hectares of land in SSA is ideal for rain-fed agriculture. This suggests that Africa possesses a considerable amount of arable land that, if improved, may ensure the region’s long-term food security. However, despite Africa’s vast agricultural potential, it has struggled to embrace modern agricultural techniques fully. If Africa does not adopt modern farming techniques, it will waste the rich resources already in the continent and be forced to rely on expensive imports [40] (Juma, 2015). It is projected that by 2025, the annual cost of importing food will increase from its current level of USD 35 billion to USD 110 billion.

**Figure 3 ijerph-19-14836-f003:**
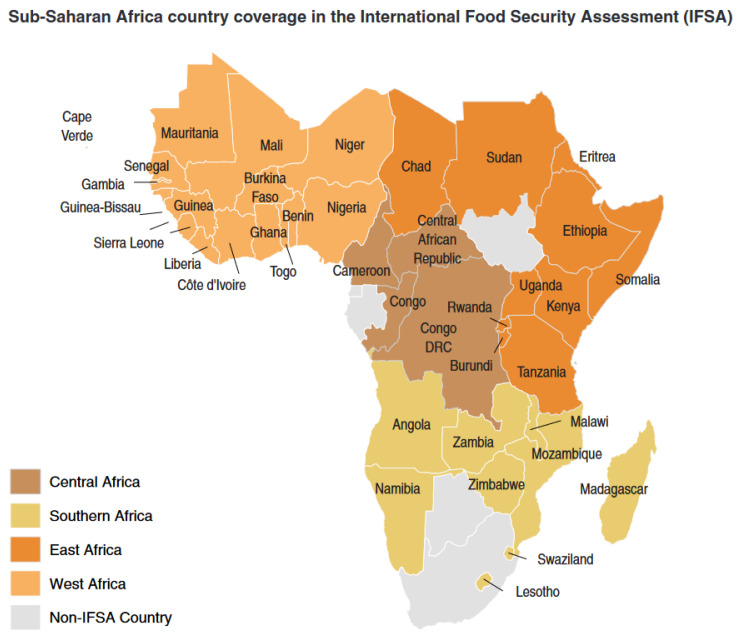
Map of the Sub-Saharan African countries. Source: Adapted from [41]. Accessed on 10th August 2022.

Currently, 346 million people in Africa are undernourished (Figure 4). With the percentage of people experiencing food insecurity expected to drop from 40.5 percent in 2020 to 24.4 percent in 2030, SSA is predicted to experience the slowest improvement in food security [42]. The World Bank reported in 2021 that 7.2 million people in East Africa are at risk of hunger, while 26.5 million experience severe food insecurity. Acute malnutrition affects at least 12.8 million children in this area.

More than 50 million people in Africa require emergency food aid annually, making it the continent that now gets the most food aid. Sixty percent of WFP’s efforts are focused on Africa [43]. Twenty-seven million people in West Africa are experiencing severe hunger, the highest number in ten years [44]. Allegedly, food crises have increased across most of West Africa, including Burkina Faso, Niger, Chad, Mali, and Nigeria [44]. Approximately 27 million more individuals have required emergency food aid from 2015–2022 [45].

Due to an unprecedented drought and the continuous conflict in East Africa, particularly in Somalia, Kenya, and Ethiopia, an estimated 81.6 million IDPs, refugees are residing in rural and urban areas within and outside these countries [46]. Southern Ethiopia, eastern and northern Kenya, and southern and central Somalia are all experiencing worsening drought conditions, which have led to a lack of water and pasture, the death of livestock, below-average harvests (65% in Somalia and 70% in Kenya’s marginal agricultural areas), a surge in the price of cereal and other staple foods, and a decrease in the purchasing power of the residents [47].

**Figure 4 ijerph-19-14836-f004:**
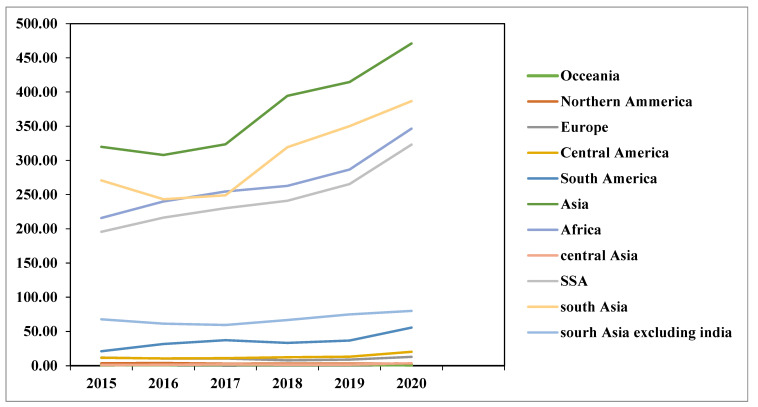
Number of severely food insecure people (in millions) by region from 2014 to 2020. Source: [48]. https://www.who.int/news/item/06-07-2022-un-report--global-hunger-numbers-rose-to-as-many-as-828-million-in-2021. Accessed on 12 June 2021.

According to [49], since 2015, there has been an increase in food insecurity across Africa. In 2015, there were 52.3 million people who experienced severe to moderate food insecurity; by 2017 that number rose to 56.7 (Figure 5). In 2020, around 66.2% of people (Figure 5) had moderate to severe food insecurity; of this number, 30% had severe food insecurity and almost 37% had moderate food insecurity [50]. Wang, Andrée, Chamorro, and Spencer [51], estimated that 704 million people living south of the Sahara experienced food insecurity in 2021.

In industrialized nations, undernourishment is less than 2.5%, but in most of Africa, the situation is dire [52]. In 2019, Figure 6 depicts the percentage of people in several Sub-Saharan nations experiencing moderate or severe food insecurity. As shown by the figure, more than half of the population in most SSA nations suffers from moderate or severe food insecurity.

## 3. Why Is Sub-Saharan Africa the Worst in Terms of Food Insecurity?

The high food insecurity rate in Africa is due to many factors, including low agricultural productivity, harsh climatic conditions, slow economic growth, poor governance, a dearth of research and innovation, and many other factors.

### 3.1. Weak Economic Growth

Throughout evolution, food security and economic development have worked together to reinforce one another. Therefore, economic expansion is the most effective tool for combating poverty, increasing food security, and enhancing the quality of life in underdeveloped nations [53]. The best and surest strategy to fight hunger and poverty is to increase economic growth [54]. A leading study of 14 nations in the 1990s found that while poverty decreased in 11 countries with high economic growth, it increased in 3 countries with low or stagnant growth. Chambo [55] claimed that an increase of one percent in per capita income, was associated with a 1.7 percent drop in poverty rates. However, Sub-Saharan Africa (SSA) has the world’s most volatile economic growth rate. From the early to the mid-1980s and between 1990 and 1994, the average annual growth rate of per capita real GDP in SSA was negative, only to rebound to 1.2 percent between 1995 and 1997 (Figure 7). Sub-Saharan Africa still has a long way to go to recover economically and socially, and the region is highly vulnerable to internal and external shocks [56]. Poverty and food insecurity persist over much of the continent despite recent improvements in economic growth rates.

The countries of Sub-Saharan Africa face formidable obstacles on their path to higher growth and greater global economic participation. However, the current economic growth rates are still insufficient to significantly reduce extreme poverty and allow these countries to catch up to other developing nations. In alleviating poverty and raising living standards, these nations need to see actual per capita GDP growth rates rise steadily and significantly [57].

Current projections indicate that many countries’ GDP per capita will not recover to pre-crisis levels until 2025 [10]. Expectations are that the growth gap between Sub-Saharan Africa and the rest of the world will worsen during the next five years. Sub-Saharan Africa is forecast to see a slower economic recovery than the rest of the world, with cumulative per capita GDP growth for the 2020–2025 period predicted at 3.6%. This is significantly lower than the projected growth rate for the rest of the world (14%) [58].

### 3.2. Gender Inequality

Gender inequality has a negative effect on achieving food security [59]. This inequality is true not only in society, communities, nations, or continents, but also in the SDGs which reflect some form of inequality in their coverage. For example, some SDGs (13, 14, and 15) that indirectly deal with food security do not incorporate gender equity or recognize how vulnerable women are, these could hinder not only the goal of food security, but also the goal of conserving resources [60].

In SSA, one problem with food security is that men and women are not treated equally. Allendorf [61] found that women owned less than 15% of farmland in SSA and less than 5% of farmland in North Africa. A study by [62] claimed that the lack of equality between men and women hurts the food security in Sub-Saharan Africa. He also said that the ratio of women to men in the work force and the gender equality in schools have a negative effect on the number of people who are undernourished and how much food security there is in the region. In the same way, Agarwal [60] in his study on gender equality and food security for the SDG found that only 10% of families in Ghana and 5% of registered landholders in Kenya are led by women. Doss et al. [63] looked at data from ten African countries: Zimbabwe, Tanzania, Lesotho, Burundi, Burkina Faso, Uganda, Ethiopia, Malawi, Senegal, and Rwanda. The results show that only 12% of the women own land as a sole ownership, even though 39% of women own land as groups. Based on the FAO’s Food Insecurity Experience Scale Survey of 2014–15, UN Women found that in two-thirds of 141 countries, more women than men say they have experienced food insecurity, with Sub-Saharan African women being the most at risk [59]. In the same way, gender is a key factor in land ownership in Nigeria. The Food and Agriculture Organization says that women farmers produce about 45 percent of the food in the country. However, land rights discrimination caused by social, economic, and cultural factors affects their productivity [64].

### 3.3. Food Price Inflation

Although a high inflation rate impacts negatively on the GDP, because it increases the cost of borrowing and slows the rate of capital investment, this effect is unlikely at low, single-digit inflation levels. Nevertheless, investors and business owners may hesitate to put money into new ventures if they cannot precisely predict expenses and earnings due to excessive inflation [65] Many empirical studies found that price shock accelerates food insecurity and poverty across low income countries. For example, ref. [66] found that the prevalence of underweight children in Mozambique increased when food prices rose. In their study of the short- and long-term effects of food price changes on poverty, ref. [67] noted that a 50 percent spike in food prices might result in a 5.8 percent increase in world poverty, and that increasing the price shock from 50 to 100 percent could double the estimated global poverty estimate to 13 percentage points. Verpooten et al. [68] investigated how the global food price shock of 2005–2008 impacted household food security status using self-reported data from over 50,000 individuals in eighteen Sub-Saharan African nations. They noticed that this adverse occurrence had a positive impact on rural families, but a negative impact on urban households. In particular, the authors noticed that the severity of food insecurity during the global food crisis decreased by 9.2 percent in rural areas from 2005 to 2008, but increased drastically by 7.8 percent in urban areas. Similarly, ref. [69] proved that not all food price increases in Ethiopia reduced household security. They discovered that the fluctuating price of teff caused the destitute people in Ethiopia to reduce their daily meal count and switch to less-desirable options. Additional research revealed that poor urban people were substantially more likely to skip or miss meals than their rural counterparts.

Recently, food prices have increased by double digits on average throughout SSA countries, with certain countries (Sudan and Zimbabwe) seeing increases of more than 180 percent (Figure 8). It is concerning because, since August 2022, the inflation rate in Ethiopia, Ghana, and Nigeria has been over 20% (Figure 8).

From June through to August of 2022, rising food prices and prolonged conflict are considered to be the cause of food insecurity for more than 38 million people in West and Central Africa [70]. Due to below-average outputs in 2021/22 and disrupted trade flows in West Africa, rising food prices are at or near record highs, especially for staple products [71]. Sixty-four percent, or 6.4 billion people, of the world’s impoverished, live in rural areas, where life is more complex [72]. For example, the FAO lists more than 30 nations with food costs: Burundi, the Central African Republic, Chad, Côte d’Ivoire, the Democratic Republic of the Congo, Eritrea, Ethiopia, Ghana, Guinea-Bissau, Kenya, Lesotho, and Liberia.

### 3.4. Low Crop Productivity

Sub-Saharan Africa’s (SSA) agricultural sector has been underperforming for some time [73]. From the early 1960s to the 1970s and up to the present, SSA’s agricultural production per capita increased by a modest amount (Figure 9). Contrasting with other developing regions, however, the overall trend is a stagnating growth [74].

SSA countries spent about USD 43 billion in 2019 for food imports [75]. Experts have been puzzled about Africa’s growing inability to feed itself despite possessing most of the world’s uncultivated farmland for a significant period of time [76]. The majority of SSA’s net agricultural import bill comes from four countries: Nigeria, Angola, the DRC, and Somalia [77]. Most of the food that is imported may be grown domestically, according to a report by [78]. This would provide much-needed employment and income for the country’s large population of unemployed young people and smallholder farmers. The African Development Bank [79] estimated that by 2030, Africa’s food import cost would be $90 billion.

### 3.5. Drought

Drought is still a problem in many African countries due to unfavorable weather patterns and climatic changes that affect rural households and agricultural outputs [80]. Droughts have killed more people than any other natural disaster since the turn of the century and have affected more than twice as many people worldwide [12]. Over 80% of Niger’s workforce is employed in agriculture, and many of the poorest people in the country live in rural areas vulnerable to unpredictable weather patterns and frequent droughts [80]. This causes perpetual production shortfalls that worsen poverty and increase food insecurity. According to [80], agricultural production is reduced due to drought, which slows economic growth, lowers employment, threatens food security, and exacerbates poverty. It has been noted by [81], for example, that communities in rural areas that rely on subsistence economies are particularly vulnerable to the effects of unpredictable rainfall on their food supply. In South Africa, for instance, farmers have lost more than ZAR 5 billion (USD 276 million) between 2014 and 2016 due to the destructive impact of drought on agricultural production [82].

Consequently, civil society groups advocating for food justice and the right to food have staged protests in response to food poverty [12]. As stated by [83], drought was the primary factor in the rapid rise in food prices and the heavy reliance on imports. The most vulnerable and impoverished populations felt the effects of this most acutely.

After four consecutive failed rainy seasons in some regions of Ethiopia, Kenya, and Somalia, food insecurity has become a severe problem in the Horn of Africa, a situation not seen for at least 40 years. The drought has impacted at least 19.4 million people in the Horn of Africa since it began in October 2020. High rates of acute food insecurity and growing malnutrition affect at least 18.6 million people every day in Ethiopia, Kenya, and Somalia, and this number could reach 20 million by the end of 2022 [84]. There are now 7.1 million people in Somalia who are severely hungry, and 213 thousand people who live in the catastrophic region of the country. Over 1.5 million animals in Kenya, 2.1–2.5 million in south and south-eastern Ethiopia, and 3 million animals in Somalia have died due to the drought. These make the Pastoralist families who rely on these animals for food and income more vulnerable to food insecurity [84].

### 3.6. Low Investment in Irrigation Agriculture and Research

The potential for irrigation agriculture to improve agricultural output, income, and food security in Sub-Saharan African countries is substantial [85,86,87]. However, the SSA countries cannot reach their full potential because of decades of underinvestment, underuse, and neglect of their public infrastructure. Only 5% of irrigable land in SSA is fitted compared to 37% in southern and eastern Asia. Comparatively, the average equipped area in SSA countries is around 450 times lower than the global average (18%) (Figure 10).

The World Bank estimated in 2000 that 85–90% of agriculture in Sub-Saharan Africa relied on rainwater for survival and that agriculture accounted for 35% of the region’s GDP, 40% of exports, and 70% of its employment [9]. Agricultural output may not be sustainable with an annual rainfall of less than 700 mm. While global scientific investments and the number of scientists have expanded in the last five years, no African country now devotes 1% of its GDP to R&D. By the end of 2022, global research investment is expected to average 2.06 percent of the GDP (Figure 11). A report by [88] indicated that 80% of the countries that invested less than 1% of their GDP in scientific research are in Africa, suggesting that poor spending on scientific research was widespread across the continent. Africa’s share of global R&D spending remained at 1.01% from 2014 to 2018 and from 2019 to 2021. In Sub-Saharan Africa, on the other hand, R&D spending accounted for only 0.45% of the GDP (Figure 11).

### 3.7. Climate Change

The African continent is the most at risk from the effects of global warming due to its poorer financial, scientific, and technological capabilities to adapt to the dangers posed by climate change and its reliance on climate-sensitive and fragile economic sectors, (e.g., rain-fed agriculture). Many of Africa’s poorest countries rely heavily on agriculture for their economies, and climate change is widely recognized as one of the greatest threats to agricultural output and food security in the 21st century [14]. Some parts of Africa (the Sahel, for example), have been drier over the last century, and it is predicted that the continent will face a steeper rise in temperature than the rest of the world [90] Considering that around 96% of food production in Sub-Saharan Africa relies on rain for irrigation, this region is very susceptible to the effects of climate change [89]. Temperature increases of 1 °C have been linked to a 2.66% decrease in agricultural output in SSA nations [91], which in turn has been shown to reduce economic growth by an average of 1.3 percentage points for each degree of warming.

Regular climate variability is seen as a threat and a barrier to sustainable global growth and development by development practitioners and policymakers. More recent data shows a concerning increase in worldwide hunger and malnutrition, along with rising instances of extreme weather [92]. Some African countries, for instance, could see a 50% drop in crop productivity by 2030 and a 90% drop in crop income by 2100 as a result of significant climate change and variability [92]. Other research indicates that national and household food security in underdeveloped nations has already decreased as a result of climate change [90]. It is possible that climatic extremes would affect the availability of food in Africa by lowering the area of viable arable land appropriate for crop production [93]. This could lead to an increase in the incidence and number of undernourished individuals in Africa. Temperature increases of 4 °C or greater, as described by [92], may have catastrophic consequences for the way of life of many farmers in Africa.

### 3.8. High Population Growth

Current estimates put the global population growth rate at 1.1% per year, with a medium-variant projection predicting that number to rise to 9.7 billion by 2050 [94]. Even though population forecasts are inherently unreliable, with recent years overestimating population growth; by 2050, the world’s population is expected to have increased to between 9.4 and 10.1 billion [93]. Sub-Saharan Africa is estimated to add 1.05 billion people between 2019 and 2050, which accounts for more than half of the global increase [95].

The 1960 estimate for Africa’s population was 257 million; by 1983, that number had more than doubled to 482 million. The continent’s population peaked in 1993 at an estimated 682 million people. However, the current SSA population is around 1.4 billion, with an average growth rate of 3.2%, which is the highest in the world. According to the prediction, Africa’s total population would reach approximately 2.5 billion by 2050 [96]. In the year 2020, the continent had roughly 1.34 billion inhabitants (Figure 12). A five-fold rise in population is predicted for several of Africa’s poorest countries, including Angola, Burundi, the DRC, Malawi, Mali, Niger, Somalia, Uganda, Tanzania, and Zambia, between 2015 and 2100 [95]. High population growth in certain regions in Africa will make it harder to eliminate problems of food insecurity, inequality, hunger, and malnutrition [97]. The growing human population and its effect on hunger have been the topic of extensive study. For instance, Molotoks et al. [98] found that food insecurity was more prevalent in nations with larger populations. The authors argued that a rising global population is a significant factor in spreading hunger throughout the world’s poorest regions, especially in Sub-Saharan Africa and South Asia.

There is a snowball effect when it comes to the costs associated with rapid population growth, as follows. (1) The current rate of births necessitates massive increases in food supply and agricultural production to keep up with the demands of a fast-expanding population, putting pressure on the government and private institutions to divert funds away from other areas of the economy and social life. (2) A higher dependency ratio is expected due to the rapid rise in the world’s population. (3) The availability of productive jobs is threatened by population growth. An increase in the available workforce typically follows a period of strong population growth, however, given that an ever-increasing number of Africans cannot find employment in the developed economies’ manufacturing and service sectors, they are either relegated to low-paying service jobs or forced back into the traditional economy’s subsistence level of production. A vast pool of potential low-wage workers can hinder technical progress, whereas widespread poverty can slow industrialization by dampening consumer demand. The ultimate effect is low saving rates and low levels of skills labor, which prevent some African countries from using the full extent of their natural riches [66].

Due to the increased need for government services in health, education, welfare, and other areas, large groups of young people, especially those unemployed or who see little hope for a satisfying future, might become a destructive and ultimately combustible political force. Since Africa’s population has boomed in the previous two decades, but its GDP per capita has been stagnant, the continent’s economic change cannot last. As a result, poverty traps become ever more entrenched [99].

### 3.9. Conflicts

According to a study by [100], conflict and insurgency are significant causes of food insecurity in Africa. The authors of [36] reported that conflict is a key contributor to food insecurity for many of the region’s countries, including Burkina Faso, Cameroon, Central African Republic, Chad, Democratic Republic of the Congo, Ethiopia, Mali, Niger, Nigeria, Rwanda, Somalia, South Sudan, and Uganda.

Grebmer et al. [101] estimated that about USD 52 billion, or 75% of total public aid received, was lost to battles in the agricultural sector between 1970 and 1997 in the SSA. Approximately 79% of the world’s 155 million stunted children and 60% of the world’s 815 million undernourished people are located in nations experiencing some form of armed conflict in the SSA [102]. The United Nations reported in February 2021 that more than 34 million people in SSA countries undergoing or recovering from violence needed food and other emergency humanitarian assistance [103]. Moreover, there are roughly 18 African countries that have been severely impacted by war and other forms of violence, such as community clashes, civil war, terrorism, banditry, political instability, and arm robbery, all of which have contributed to widespread poverty and food insecurity in the region [9].

According to a study by [104], conflict has had a significant and long-lasting effect on the economy of Sub-Saharan Africa. They also asserted that the cumulative negative effect of conflicts on per capita GDP increases over time, leading to an increase in the already ubiquitous rate of food insecurity. Their findings showed that the GDP of the countries in conflict could shrink by 2.5 percent. Bircan et al. [105] reported that the number of people who have been compelled to leave their homes as a result of a crisis in SSA countries has increased considerably from approximately 2 million in the early 2000s to 10 million in 2017. The major countries on the list are: the Democratic Republic of the Congo (4.4 million), South Sudan (1.9 million), and Nigeria with around 1.7 million people.

In Nigeria and other parts of West Africa, conflicts between herders and farmers are a serious problem [9]. Bandits and other criminals have forced hundreds of thousands of farmers to abandon their rural or farm communities. Attacks by Boko Haram and the Islamic State in West Africa have forced 2.5 million people to flee their homes in Nigeria’s north east [9]. Eight hundred thousand more people have been forced to leave their homes in the northwest because of kidnappings, extortion, and attacks by organized criminal groups [106]. A study by [107] estimated that Africa lost over 120 billion USD in agricultural output due to wars between 2016 and 2017. Border violence between Nigeria and Cameroon poses a serious risk to the food security situation of both nations. The Ambazonian separatist struggle has prompted Cameroonians to migrate into Nigeria, while the Boko Haram insurgency in the North-East has caused internal population displacement and pushed many to become refugees in Cameroon, Chad, and Niger [104]. About 2.3 million people’s lives are in jeopardy because of the crisis in Cameroon, and more than 4.3 million people need immediate access to food, water, shelter, and healthcare, as reported by [108].

### 3.10. Corruption

In every culture, corruption signifies a more profound and widespread sickness. Although it is not exclusive to any country or region, it is the most widespread in Sub-Saharan Africa [72]. Evidence from Transparency International showed that only seven Sub-Saharan African (SSA) nations achieved a higher than average Corruption Perceptions Index (CPI) score in 2019 [72]. Bribery, like other forms of economic instability, has the effect of widening the income gap between the rich and the poor and fueling widespread poverty and food insecurity in SSA [109]. Sub-Saharan Africa has some of the highest rates of corruption in the world, according to [110]. Corruption and the misuse of public finances have exacerbated social division and armed conflict in several SSA nations [111]. While many SSA governments claim to be concerned about ending malnutrition, few have taken the time to quantify the extent of the problem. Mbate [112] in his study on the link between corruption and food insecurity, claimed that poor rural people in the SSA are especially susceptible to corrupt practices. He further claimed that impoverished people in Kenya were shown to pay bribes more frequently and with a greater proportion of their income than the middle class and upper class, thus, pushing them into a more devastating state of food insecurity. Anser et al. [113] discovered that in the West African sub-region, poor governance and the high incidence of corruptions could reduce food security by 20%. In Malawi, urban residents overwhelmingly identified the Cashgate scandal as the root cause of household poverty and food insecurity [114].

### 3.11. Other Factors

According to [24], social and economic issues are the primary causes of food insecurity among African families. This research showed that the significant income earner’s education, income, and wealth are the critical factors for food security. Some studies [115,116] argue that extreme poverty poses the greatest danger to food security in Africa. Inadequate agricultural policies, poor infrastructure and transportation, insufficient marketing strategies, frequent extreme weather conditions, a rising disease burden, a weak financial support scheme, insufficient safety net systems, and political conflicts were all cited as contributing to the current food insecurity in Africa. According to [117], food insecurity in Africa is multifaceted and has various origins. Healthcare, disputes, policies, politics, leadership, strategic vision, economic interests, agricultural production, the food system, global food industry commerce, and environmental sustainability are a few of the domains in which this problem manifests. The following are considered to be significant factors contributing to chronic food insecurity in African countries: (1) constant political instability and crises; (2) brief or protracted civil conflicts and wars; (3) pervasive, persistent, and institutional corruption; (4) incorrect economic policies and mismanagement; (5) a lack of committed political leadership; and (6) a lack of clear financial and economic investment in the region [72]. The relationship between bribery at work and food insecurity in SSA was studied by [71] who discovered that those who had been bribed were more likely to go hungry than their non-bribed counterparts.

#### The Gateway to Sustainable Food Security in the SSA

Increasing food security calls for a varied and involved set of actions. Table 1 provides a concise summary of the most significant ones.

## 4. Conclusions

While the agricultural sector plays a critical role in enhancing food availability, guaranteeing food security has become an issue of key concern for countries in Sub-Saharan Africa with varying degrees of economic development. The aim of this paper was to investigate: (1) why SSA countries respond slowly to the SDG2 goal of “End hunger, achieve food security and improved nutrition, and promote sustainable agriculture” and also (2) to showcase strategies that Sub-Saharan African countries should adopt to improve food security in a sustainable manner. This study focuses not only on identifying the reasons for high food insecurity in SSA, but also adds to the understanding of the most effective strategies to alleviate the hunger problem under the particular characteristics of SSA. It presents a holistic lookout for the policy options in various regions across the globe, that may interest to scholars and policy makers. The results from the literature reviewed show that the greatest problems related to sustaining food security in Sub-Saharan Africa include: weak economic growth, gender inequality, high inflation, low crop productivity, low investment in irrigated agriculture and research, climate change, high population growth, poor policy frameworks, weak infrastructural development, and corruption, among other things. Promoting investments in agricultural infrastructure and extension services together with implementing policies targeted at enhancing the households’ purchasing power, especially those in rural regions, appear to be the essential drivers for improving both food availability and food access.

This study is not without limitations. First, the study was limited to Sub-Saharan Africa, which shares similar characteristics to South East Asia; therefore, there could be other factors responsible for food insecurity which are not captured in this review. Secondly, it should be stressed that the findings of this study do not constitute an all-inclusive evaluation of food security. Future research should incorporate other factors to have a comprehensive assessment of food security in Africa.

## Figures and Tables

**Figure 1 ijerph-19-14836-f001:**
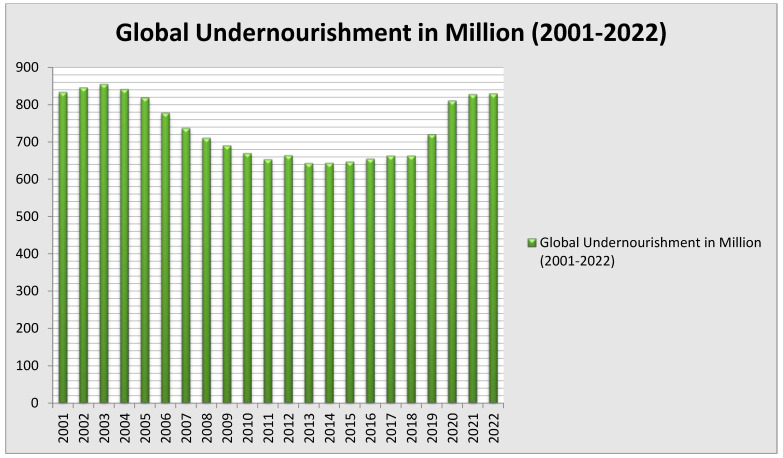
Global undernourishment in millions. Source: World Bank Database, 2022.

**Figure 2 ijerph-19-14836-f002:**
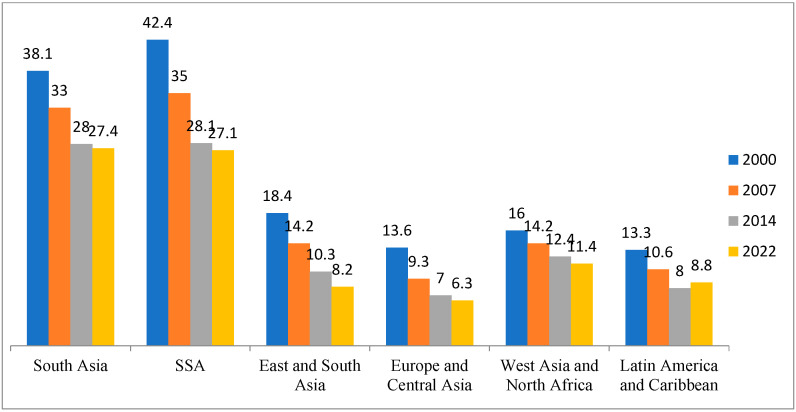
Regional Global Hunger Indexes in 2000 to 2022. Source: https://www.globalhungerindex.org/trends.html, accessed on 3 October 2022.

**Figure 5 ijerph-19-14836-f005:**
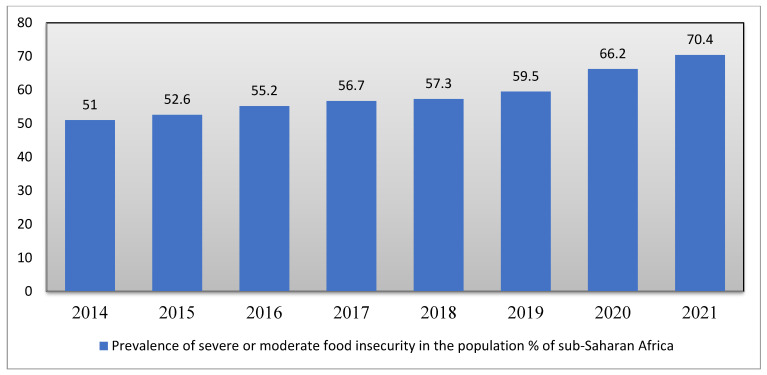
Prevalence of severe or moderate food insecurity in the percentage of the population of SSA. Source: https://data.worldbank.org/indicator/SN.ITK.MSFI.ZS?locations=ZG. Accessed on 12 June 2022.

**Figure 6 ijerph-19-14836-f006:**
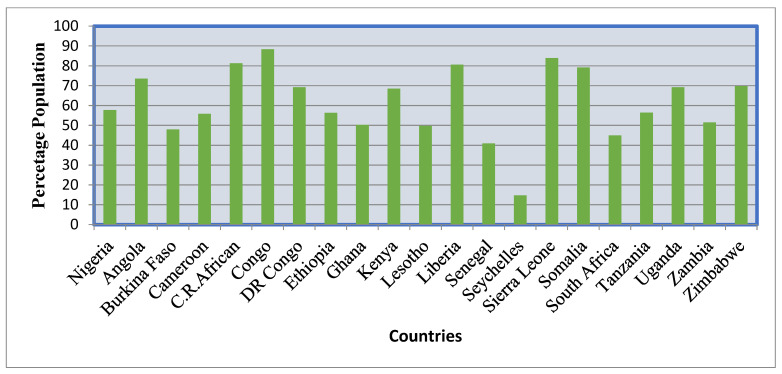
Prevalence of moderate or severe food insecurity in the percentage of the population of some Sub-Saharan countries in 2019. Source: https://data.worldbank.org/indicator/SN.ITK.MSFI.ZS?locations=ZG. Accessed on 25 May 2022.

**Figure 7 ijerph-19-14836-f007:**
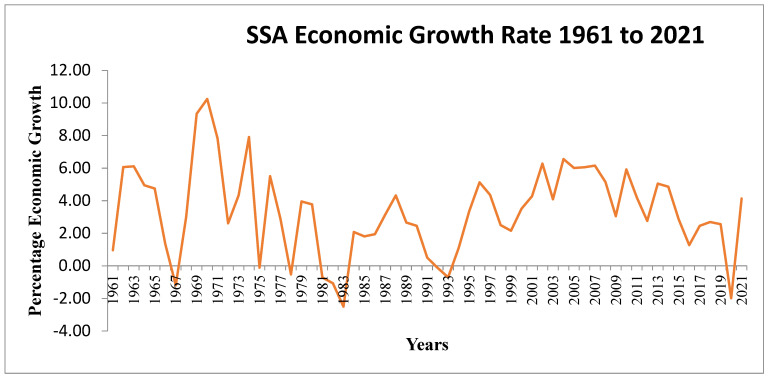
Sub-Saharan African economic growth rates over time. Source: Macrotrends, 2022.

**Figure 8 ijerph-19-14836-f008:**
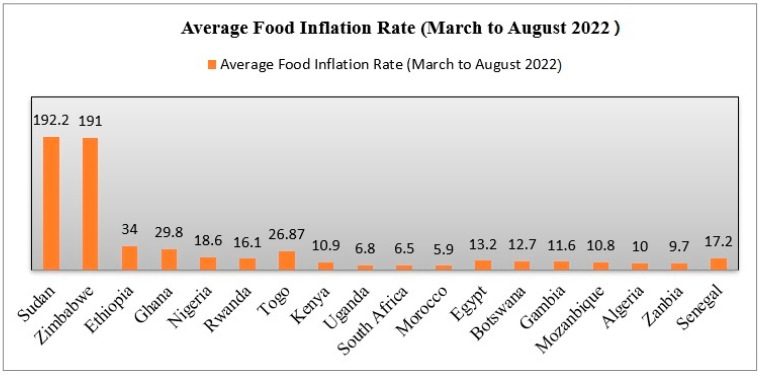
Average inflation rates of some African countries (March to August 2022). Source: Trading economics, 2022.

**Figure 9 ijerph-19-14836-f009:**
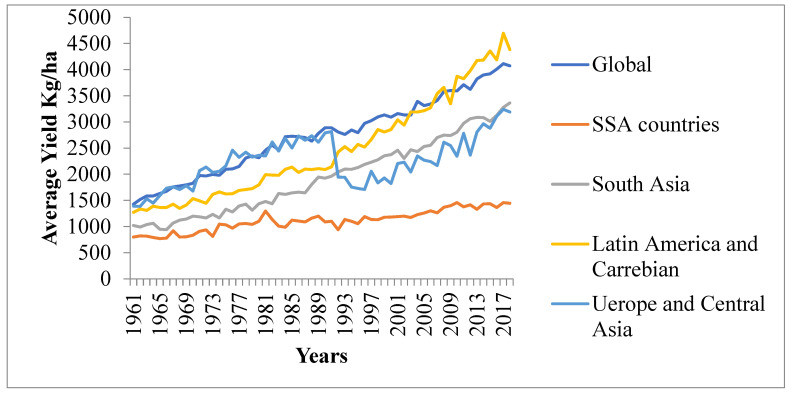
Average yield of cereal from 1961 to 2018 for different regions and the global average. Source: https://data.worldbank.org/indicator/AG.YLD.CREL.KG?locations=ZG. Accessed on 22 February 2022.

**Figure 10 ijerph-19-14836-f010:**
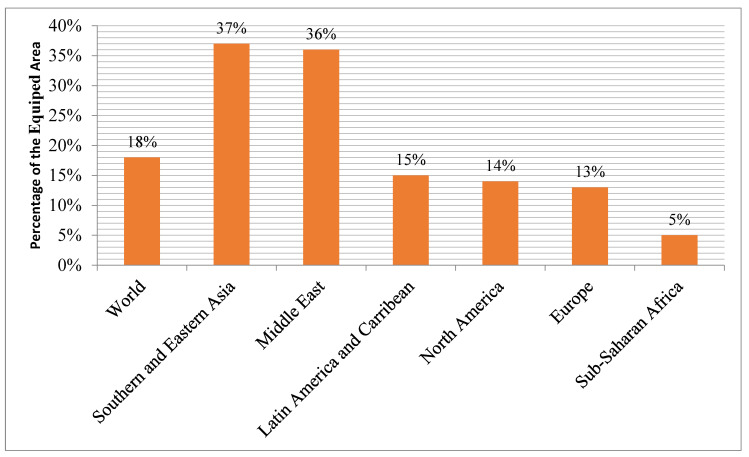
The equipped area under irrigation by region. Source: Domenech, 2010.

**Figure 11 ijerph-19-14836-f011:**
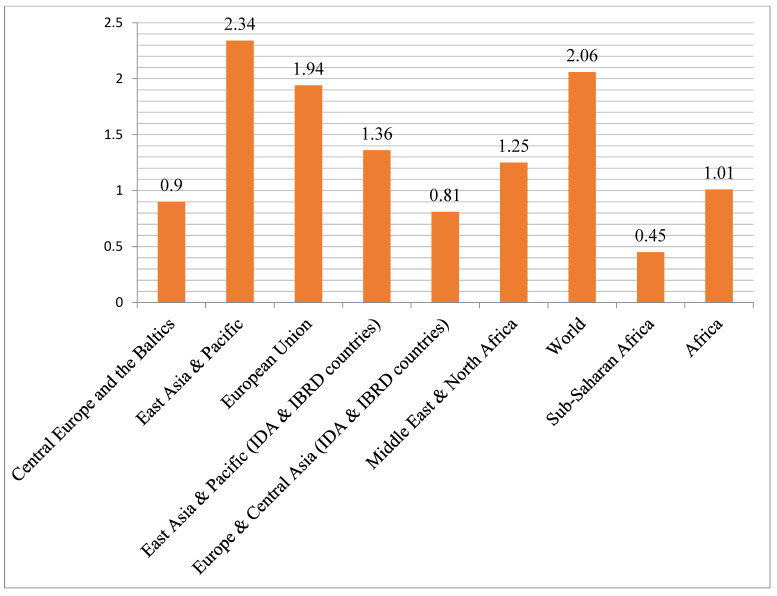
Research and development expenditure (% of the GDP)—Sub-Saharan Africa by countries from 1980–2021. Source: [89].

**Figure 12 ijerph-19-14836-f012:**
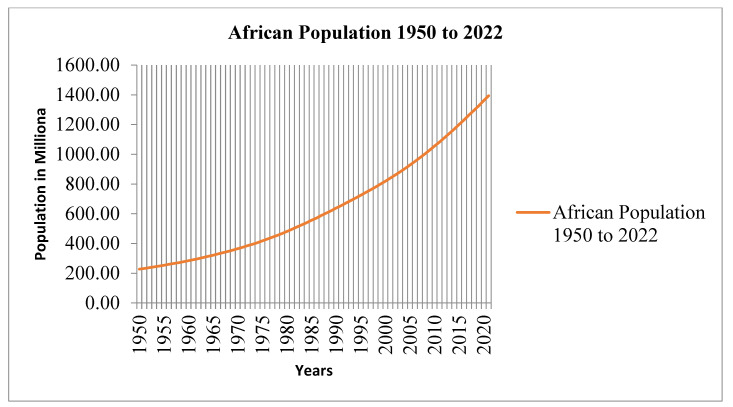
The African population from 1950 to 2021. Source: https://www.macrotrends.net/countries/AFR/africa/population: United Nations world population prospect. Accessed on March 2022.

**Table 1 ijerph-19-14836-t001:** The Gateway to Sustainable Food Security in the SSA Caption.

S/N	Policy Option	References
1.	Effective agricultural policies, provision of intermediate technology, women’s empowerment and social protection, farmers’ training, and adoption of a participatory approaches in agricultural projects and programs	[9,11]
2.	Investments in irrigation and rural infrastructures for sustainable development and economic growth can improve food security at the regional or national level.	[116,118,119,120]
3.	Transformation of the entire economy from a consumer-driven economy to a productive economy by enhancing productivity growth and regional trade.	[9,75]
4.	Research and innovation in agriculture and entrepreneurship, as well as changing the direction of our education system toward skill acquisition	[121,122]
5.	Measures to reduce income inequality, corruption, and poor governance	[123,124]
6.	Improvement in macroeconomic policies, economic diversification, and structural reform, including privatization programs and social and human resource development.	[125,126]

## Data Availability

Data may be requested from A.A.H.

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
