# Peer review of "Reversing Years for Global Food Security: A Review of the Food Security Situation in Sub-Saharan Africa (SSA)"

_ijerph, 2022, doi:10.3390/ijerph192214836_

Round 1

Reviewer 1 Report

Article Type: Review article
Title: Reversing Years for Global Food Security: Review of the Food Security Situation of the Sub-Saharan Africa (SSA)

Review report

This paper examines the food security situation in Sub-Saharan Africa (SSA) and discusses the multidimensional obstacles to achieving SDG2 goals and food security status in SSA regions. The topic of the manuscript is relevant to the International Journal of Environmental Research and Public Health. It analyzed the major barriers to sustaining food security in Sub-Saharan Africa; the methodology of the study they discussed is very clearly explained. However, the following issues should be addressed in order to improve the manuscript further.

Concerns

1. The Introduction section, particularly the Background section, should provide detailed discussion so that the significance of Sub-Saharan Africa's (SSA) food security situation can be highlighted. The introduction section should cite more recent papers in order for readers to understand the actual food security situation in the SSA region in comparison to other underdeveloped regions of the world, such as South Asia and Southeast Asia. Therefore, authors should cite papers from other countries in addition to papers from Africa. For example, authors are advised to include the following relevant papers that have previously been published in the line number 44.

(a) Economic impact of climate change on crop farming in Bangladesh: An application of Ricardian method; (b) Revegetation of coal mine degraded arid areas: The role of a native woody species under optimum water and nutrient resources; (c) Smallholder farmers’ willingness to pay for flood insurance as climate change adaptation strategy in northern Bangladesh

Novelty must be presented more clearly, highlighted, and academic and professional applications need to be indicated.

2. Discussion of the analysis with a comparison and a proper literature review is necessary. Proper geographical distribution of the references will broaden the study and allow it to be extended to other places. Papers from different journals that approach the subject are desirable as well.

3. The discussion part also needs to underline how this can be generalized and how the validation could be achieved.

4.  The authors are also suggested to mention some limitations of this study and to make specific and solid recommendations to policymakers based on the findings of the research.

Author Response

Dear reviewer,

thank you very much for preparing the review. We agree with all comments. We improved the manuscript with all the guidelines.  Below are the changes we have made according to the reviews:

  1. Novelty of the work has been improved more pieces of information are added
  2. Noted and improved
  3. The discussion part has been improved more relevant literature were added to the manuscript
  4. Limitation of the study is added and the recommendation is improved

Thank you very much for helping us improve our manuscript. We appreciate the input and substantive comments.

Reviewer 2 Report

This review of factors contributing to food insecurity in SSA is desperately needed in the literature.  However, the authors have relied heavily on advocacy publications, opinions, and outdated data to make their case.  I would suggest a review of more current data, removal of non-empirical statements, and original data sources in the references.  This is a very important topic that deserves the greatest attention to detail and current statistics.  

While UNICEF, WHO, and the WFP are trusted and credible organizations, references to the original data reported on their websites would make this a stronger article.

Line 118 – should this be 2022 (or later) instead of 2020?

Line 122 – Remove “whopping”

Line 125 – All the change occurred in 2019?

Line 132 – Is food safety the issue?

Line 124 – The purpose of the linear line in Figure 1 is unclear.

Line 147 – Is 2015 the most current data available?

Line 154 is an incomplete sentence.

Line 172 – Sentence ends abruptly after “by.”

Line 199 – Is there supposed to be text after “to whit?”

Line 200 – “as stated” is redundant

Line 234 – Are the authors addressing food safety?

Line 241 – Remove “sadly.”

Section 3.1 is based on economic data prior to 2009 with “projections” of future expectations.  Are there more current data to strengthen this section?

Section 3.2 would be stronger if more specific to SSA food insecurity.

Line 274 – “…differences” in what?  Reference is 10 years old – has this changed?

Line 278 – what is “this trade-off?”

Line 283 – “By these” what?

Line 284 (i.e., fertilizers…)

Line 299 – Should this be “rising food costs…”

Figure 9 needs a spelling check.

Line 315 needs a reference

Line 326 – this reference is nearly two decades old – it is imperative to have more current references for this section.

Section 3.5 should be “Drought”

Line 341 – Authors may want to include the U.S. dollar equivalent for the rands since this is an international journal.

Figure 10 is based on data from 2010.  More current data should be used.

Line 374 – should be a comma after “end of 2022”

Lines 396-407 – Again, more current data should be used.

Line 413 – “will be alive in 2021…” Shouldn’t these number be predictive of the future instead of the past?

The entire 3.8 section has no supporting references. The sentence 429-431 is particularly opinionated with no empirical basis. In addition, the 439 sentence negates the entire 3.8 section.

Line 439 – this is one of the strongest (and best written) sections of the paper but is simply labelled as “other factors.”  Conflict and corruption are significant hindrances to growth in African countries and this should be a major category of concern in the paper.

Line 470 “findout”

Line 473 – needs a table listing the strategies

Line 482 – “development”

Several references are incorrectly formatted.

Author Response

Dear reviewer,

thank you very much for preparing the review. We agree with all comments. We improved the manuscript with all the guidelines. Thank you very much for helping us improve our manuscript. We appreciate the input and substantive comments. Below are the changes we have made according to the reviews:

Line number

Corrections

118

Corrected

122

Removed.

125

Observed and corrected

132

The statement is in Page 134 and is removed

124

The line has been removed

147

Dully observed and adjusted

154

Dully observed and revised

172

Corrected

199-203

Modified

200

Modified

234

The line number is 236, observed and corrected

241

removed

Section 3.2

Corrected and specific to SSA

274

All the section has been recast

278

corrected

283

The entire sentence has been re-cast

299

corrected

Figure 9

315

Reference inserted

326

The statement has been removed

Section 3.5

Corrected

341

Dollar equivalent inserted

Fig. 10

Recent data for equipped area under irrigation in Africa could not found

374

Comma inserted

396-407

The entire section has been recast.

413

Adjusted

439

Revised, separate subsection  for conflicts and corruption were made

470

corrected

473

Table created

470

Adjusted

Round 2

Reviewer 2 Report

There are multiples errors in grammar throughout the text – please proofread and revise